# The Effect of Choice Attributes of Internet Specialized Banks on Integrated Loyalty: The Moderating Effect of Gender

**Ji-Hee Jung [1] and Jae-Ik Shin [2],\***

[1]  Department of Business Administration, University of Ulsan, 93 Daehak-ro, Nam-gu, Ulsan 44610, Korea; aboutjee@ulsan.ac.kr

[2]  Department of Distribution, Gyeongnam National University of Science and Technology, 33 Dongjin-Ro, Jinju, Gyeongnam 52725, Korea

\*   Correspondence: sji@gntech.ac.kr; Tel.: +82-55-751-3663

**Abstract:** There is growing interest in how Internet specialized banks that provide the Internet as a major customer channel can change the paradigm of the banking industry by securing a niche market in competition with existing banks that have a multi-channel strategy. The purpose of this study is to investigate how the choice attributes of Internet specialized banks affect attitudinal loyalty and intention of continuous use, and to identify the moderating effect of gender. A structural equation analysis was performed using 215 respondents. They have experience using Internet specialized banks in South Korea. The results of the empirical analysis are as follows. First, information, transaction, and safety of choice attributes in Internet specialized banks have a positive effect on attitudinal loyalty. Second, information, transaction, and safety of choice attributes in Internet specialized banks have a positive effect on intention of continuous use. Third, attitudinal loyalty has a positive effect on intention of continuous use. Fourth, the gender moderating effects between information, transaction, safety and attitudinal loyalty, and intention of continuous use are rejected at the significance level of 0.05. It is found that information, transaction, and safety, which are choice attributes of Internet specialized banks, are the main factors that improve attitudinal loyalty and intention of continuous use. Contrary to expectations, the gender moderating effect between the choice attributes, attitudinal loyalty, and intention of continuous use is not significant, but there is a difference in the degree of influence between men and women. Therefore, in order to improve the performances of attitudinal loyalty and intention of continuous use in Internet specialized banks, the choice attribute should be managed from users' perspective. The management of successful choice attributes that customers want will be the foundation for Internet specialized banks' sustainability.

**Keywords:** Internet specialized bank; choice attribute; attitudinal loyalty; intention of continuous use; sustainability

## 1. Introduction

In April 2017, Korea's first Internet specialized bank, K-Bank, was launched, and in July 2017, Kakao Bank was launched. The two banks have shown remarkable growth since their launch, and as of September 2018, the total assets of domestic Internet banks amounted to 17.7 trillion won (14.5 trillion won for Kakao Bank and 2.5 trillion won for K-bank). The Internet specialized bank refers to a bank without physical stores or very few branches, and most of its operations are conducted through electronic media such as Automated Teller Machin (ATM) and the Internet. In particular, domestic Internet specialized banks have provided savings/loans interest rates to consumers through launching and growth, and have promoted competition in the domestic banking industry. The companies have

contributed to increasing their users' convenience by expanding non-face-to-face transaction services and strengthening Internet/mobile channels. New competitors, which have not appeared for 24 years, have entered the banking industry to strengthen consumer benefits and services, and contribute to strengthening the competitiveness of domestic banks. There is a growing expectation for Internet specialized banks to bring about this change.

According to Nielsen Korean Click, the Kakao Bank app monthly users (MAU) recorded 703 million in June 2019. MAU is the number of users who access the app at least once a month. This is the first time Kakao Bank has surpassed all commercial banks in terms of MAU. About 70% of Kakao Bank's 10 million customers use the app at least once a month. Kakao Bank has the largest number of installed mobile devices, with 8,807,230 units, Kookmin Bank with 8,071,989 units, NH Nonghyup Bank with 759,007 units, and Shinhan Bank with 6,679,332 units. Three out of 10 offline commercial bank app users are using Kakao Bank. Kakao Bank succeeded in turning to a surplus of 6.6 billion won in 1 year and 6 months in the first quarter of 2019, and the number of account openings exceeded 10 million in July 2019. More than one-third of the economically active population opened Kakao Bank. This is an unprecedented speed among the world's Internet specialized banks. Moreover, the fact that more than 45% of future customers in their 20s and 30s are major customers suggests that Kakao Bank will become a core bank in the next decade.

Although the competitiveness of the domestic banking industry and the user convenience of Internet banking and mobile banking have been greatly improved, many previous studies of the fast growing Internet banking have focused on attracting new customers and retaining existing customers based on the theories of technology acceptance, innovation diffusion [1,2], security [3], service quality [4], motivator [5], satisfaction [6], and age segmentation [7]. These studies have not addressed the analysis on customer retention, loyalty, and continuous use of Internet banking abundantly.

Innovative mobile Fintech payment services such as Apple Pay, Samsung Pay, Kakao Pay, etc., are driving the mobile payment sector from the consumers' perspective [8]. Since its eventual success is determined primarily by users' continued use, initial adoption of Fintech payment services is a crucial first step [9]. The success of Internet specialized banks based on Fintech will also depend on their customers' loyalty. Thus, a research on integrated loyalty is needed in the Internet specialized bank sector.

In recent years, many firms have strengthened their customer service to meet customers' needs and satisfaction for building a competitive advantage [10]. In order to provide more efficient and accessible financial services, financial technology (Fintech) services combine finance and technology [11]. There has been a study on the relationship between service attributes and behavioral loyalty in other services sector [12]. They used emotion, ambiance, and staff attitude as service attributes in casino services. Ahn and Lee [13] examined how economical, convenience, and emotional values affect usage intention in the Internet special bank sector. Lee [14] presented quality of information, transaction, website design, communication, and security as e-SERVQUAL when existing online service evaluation criteria. In general, product attributes have been useful in differentiation strategies. Product attributes also have a positive effect on customer loyalty [15]. Therefore, this study intends to use information, transaction, and safety as attributes to select Internet specialized bank services.

Therefore, it is important to understand what factors influence customer retention and loyalty in the recent Fintech competition market. Analyzing the economic effects of existing customers and technologies of 13 newly established Internet specialized banks in Europe, it revealed that the scale effect of technology base exists, but that the customer base should be prioritized to improve profitability [16]. It is well known that identifying and satisfying customer needs ensures long-term profitability for the organization. Internet specialized banks can only succeed if they adhere to this principle. The domestic Internet specialized bank and Fintech markets are entering the growth phase through rapid introduction. The service attributes of Internet specialized banks that have entered the growth phase after the introduction phase should be clearly perceived among consumers in Korea. It is

also necessary to study the Internet bank users to maintain and secure customer loyalty in response to market changes.

The world has unquestionably become a very different place for marketers by globalization, deregulation, market fragmentation, consumer empowerment, environmental concerns, and all the remarkable developments in communication technology. Recently, sustainability has been an important issue. There is a triple bottom line—people, planet, and profit—and the people part of the equation must come first. Sustainability means more than being eco-friendly [15]. Corporate actions toward achieving sustainability take all forms. According to Kakao Bank's Code of Ethics, it focuses on strengthening environmental protection campaigns for the sustainable development of future generations, and customized services to solve digital financial alienation of the elderly. Therefore, this study can suggest the implications of sustainability in the Internet specialized bank sector.

Therefore, this study aims to confirm the effects on the choice attributes and integrated loyalty (attitudinal loyalty and intention of continuous use) of Internet specialized banks and to suggest implications. The implications will include sustainability along with the marketing strategy of Internet specialized banks. First, this study examines the relationship between the service's choice attributes and integrated loyalty, which have not been covered in the previous studies of Internet specialized banks. This will show how the simplicity and convenience of the choice attributes in online financial transactions affect the formation of users' integrated loyalty. Second, it analyzes whether there is gender difference when the choice attributes have a positive effect on integrated loyalty. This will show whether there is a difference in gender roles in online financial transactions. It may also suggest gender implications for sustainable development. Third, the relationship between attitudinal loyalty and intention of continuous use will be identified within integrated loyalty. This will show whether the relationship between attitudinal loyalty and behavioral loyalty meets general expectations. Fourth, this study will provide academic and practical implications that contribute to the activation of Internet specialized banks.

## 2. Literature Review and Hypothesis

### 2.1. Internet Specialized Bank

Internet specialized banks refer to banks in which most of their business is conducted through electronic media such as the Internet or ATM without a physical branch. In the early days of its establishment, it became a completely non-store form and was referred to as a virtual bank, an Internet-only bank, an online-only bank, and a pure-play internet bank. Since then, more and more physical offline facilities have been supplemented, which is called the "Internet specialized bank". Internet banking and mobile banking are conceptually distinguished from the Internet specialized bank in that they do retail banking in physical facilities and provide financial services through the Internet or mobile. In particular, with the convergence of technologies such as Internet networks, smartphones, and Social Network Service (SNS), the new Fintech service paradigm is shifting, consumer's role in the service is expanding, and the manner and custom of financial business are rapidly changing [17].

Internet specialized banks are a new form of financial industry following the digital revolution of the 4th Industrial Revolution. Customers can improve convenience, accessibility, and profitability, and the government can enhance financial innovation by introducing new competition through such changes in the financial industry. Financial consumers' expectations for mobility and immediate access are growing, but traditional banks are not realistically meeting these requirements. Internet specialized banks focus on non-face-to-face channels that reflect customer convenience and accessibility needs, and are willing to reform profit-oriented structures in the traditional financial industry [18]. In addition, the expansion of non-face-to-face channels verified by various financial institutions is increasing the legitimacy of strengthening Internet banking or establishing Internet specialized banks. Due to the possibility of prolonged recession, Internet specialized banks, which can be an alternative investment source to receive higher profits than commercial banks, have been highlighted [19].

Many studies dealing with Internet specialized banks have been based on various theories such as Technology Acceptance Model (TAM), Diffusion of Innovation, Integrated Technology Acceptance Theory (UTAUT), and Socio-technical Theory. On the basis of this, the factors that play an important role in consumer acceptance of Internet specialized banks have been identified [1,2]. Delgado et al. [16] predicted that Internet specialized banks would be less profitable than traditional banks because of their high initial non-operating expenses. To analyze this, they compared the profitability of 13 new Internet specialized banks, 290 small traditional banks, and 45 new traditional banks in Europe. The result showed that Internet banks, which had a weak customer base, were less likely to have lower interest rates, resulting in lower profitability and efficiency than traditional banks. Therefore, it was confirmed that the success of Internet specialized banks is affected by securing customer base rather than technology, and the profitability improves when asset size expands and non-interest operating cost decreases.

Arnaboldi and Claeys [20] compared and analyzed the performances of traditional banks and those of Internet banking between 1995 and 2004 in order to find out why European banks, which have strong control over the banking industry in the UK, Italy, Finland, and Spain, establish Internet banking as subsidiaries of banks and insurance companies. The analysis result showed that the performance of Internet specialized banks in financial groups can be similar to that of existing banks that use Internet banking, and their profitability depends on each country's economic situation. It was confirmed that an Internet specialized bank could be established to diversify the risks of the banking group. It also suggested that the efficiency of Internet specialized banks could be the solution to solve the overbanking problem caused by excessive face-to-face services in Korea.

Sustainability ratings exist, but there is no consistent agreement about what metrics are appropriate. One comprehensive study used 11 factors to assemble a list of the top 100 sustainable corporations in the world: energy, water, $CO_2$, and waste productivity; leadership diversity; CEO-to-average-worker pay; taxes paid; sustainability leadership; sustainability pay link; innovation capacity; and transparency [15]. Internet specialized banks are also interested in environmental protection campaigns. If the environmental protection efforts of Internet specialized banks are well known among consumers, a positive image of the company will be built.

## 2.2. Choice Attribute

In order for Internet specialized banks to survive and succeed in the changing financial market environment, it should provide excellent customer value that consumers want. In addition, as the market size grows, competition is intensifying, and efforts are needed to maintain existing customers and derive successful marketing strategies. Internet specialized banks are latecomers in banking. Customer acquisition of Internet specialized banks should be accompanied by customer churns from existing market entrants. To this end, domestic Internet specialized banks should make efforts to provide differentiated services based on additional promotion strategies, as well as enhance the competitiveness of the product itself and services.

When customers perceive service quality, some attributes that they consider have a high and low level. In order to improve the quality of perceived services, it is necessary to identify the attributes that are more important to customers and to manage them [15]. Choice attributes are important when customers select goods and services and are explained as factors when they decide to spend on products.

Choice attributes are a fundamental approach to analyzing consumer behavior in that it can meet the needs and expectations of their decision-making [21]. Furthermore, choice attributes have a decisive influence on consumer's product or brand selection, which is a very important factor in establishing effective marketing [22]. Thus, choice attributes can be called the main factors to be considered when selecting Internet specialized banks to trade.

Lee [14] synthesized SERVQUAL and existing online service evaluation criteria, and conducted empirical research. The five factors of e-SERVQUAL were presented as "quality of information,

transaction, website design, communication, and security". As the scale of e-SERVQUAL was developed, it was used to establish marketing strategies for Internet services and financial services.

Companies are strategically using e-SERVQUAL to build website choice attributes. Therefore, this paper organizes the choice attributes of Internet specialized banks into three dimensions based on previous studies.

Information refers to the freshness and accuracy of product information, as well as product assortment. Therefore, Internet specialized banks have a high proportion of financial product information and can be applied to other financial products such as inquiry, transfer, loan application, and card use. Transaction includes the convenience of procedures and functions required for users to use financial services through Internet specialized banks. It also includes matters related to the installation and use of Internet specialized bank systems, and complaint handling. Safety is especially important for the use of Internet specialized banks. It means the stability of the system, the speed of use, transmission, and application installation. It also includes the protection of user privacy and transaction records. If the innovation of the Korean financial market is delayed, it is highly likely to give domestic financing markets and customers to foreign Fintech companies, so post-regulation such as monetary sanctions is required for financial oversight authorities to prevent such a negative situation. The main reasons for not using mobile payment are 'information leakage and security concerns' (78.3%), 'safety device distrust' (75.6%), and 'concern for accidental loss' (70.7%).

Liu's [23] study verified that consumption tendency positively influences the choice attribute of mobile simple payment service and the choice attribute of mobile simple payment service has a positive effect on behavioral intention. Based on the contents of previous studies, this study aims to present information, transaction, and safety as the choice attributes of Internet specialized banks. The reason for selecting the choice attribute as information, transaction, and safety is because it considers the service characteristics of Internet specialized banks. Deposit, bank transfer, loan, and simple payment services are non-face-to-face and mobile. The services are not also constrained by time and space.

## 2.3. Attitudinal Loyalty

The development of information technology, the easing of financial regulations, autonomy, and financial expansion have made the boundaries between industries insignificant, opening the way for diversification of profit sources by expanding financial products and services provided by financial companies. On the other hand, however, competition within homogeneous financial firms is expanding to competition among financial companies in different territories [24]. Therefore, this change in financial environment led to a shift to relationship marketing that strengthens the relationship with existing customers rather than creating new customers [25].

Customer retention has emerged as an important factor in corporate performance, and firms emphasize marketing activities such as loyalty management to maintain continuous customer relationships [26]. Attitudinal loyalty leads to behavioral loyalty, such as repeated purchases for a particular brand. In other words, from the perspective of customer-based brand assets, it is in line with the importance of raising the mind share of customers first, not the market share, in order to build a strong brand asset [27]. According to a conceptualization study of customer loyalty, it includes attitudinal loyalty represented by psychological obsession and behavioral intention, behavioral loyalty represented by repetitive purchasing patterns, and the composite stream represented by the actual behavior [28].

If customer loyalty does not involve a favorable attitude toward the brand or company, customers can more easily switch to another company or brand [29]. Therefore, loyalty is consistently considered as a major outcome variable in marketing and consumer behavior studies. Many loyalty studies show that attitudinal loyalty acts as a leading factor in behavioral loyalty [30]. In other words, after psychological attachment is formed to the goods and services, behavioral loyalty, such as repeated purchases and continuous use, appears [31].

Dick and Basu [32] described loyalty as a relationship between relative attitudes to particular objects and repurchase behavior, and proposed a theoretical model that considered both attitudinal and behavioral loyalty. Previous studies that considered the concept of loyalty as a two-dimensional structure have no disagreement that attitudinal loyalty is a variable that precedes behavioral loyalty. This means that the emotional attachment to the brand can appear as a repeat purchase behavior [33].

The concept of loyalty should not only have a behavioral element of 'repetitive buying behavior' but also a favorable attitude of 'emotional attachment' to the object. Regarding the structure of customer loyalty, many scholars generally support the view that customer loyalty has at least a two-dimensional structure of behavioral and attitudinal factors. Brand loyalty, a type of customer loyalty, has at least two sources: attitudinal and behavioral loyalty. Brand loyalty appears in the form of behavior, with some degree of temporal continuity [34].

Therefore, the following hypotheses are presented to examine how the choice of internet banks affects attitudinal loyalty.

**Hypothesis 1 (H1)**— *Information of choice attributes in Internet special banks will have a positive influence on attitudinal loyalty.*

**Hypothesis 2 (H2)**— *Transaction of choice attributes in Internet special banks will have a positive influence on attitudinal loyalty.*

**Hypothesis 3 (H3)**— *Safety of choice attributes in Internet special banks will have a positive influence on attitudinal loyalty.*

## 2.4. Intention of Continuous Use

Intention of continuous use is defined as the degree to which the user would like to use the Internet specialized bank service again and which it is recommended to others [35]. It is a key concept for a continuous relationship between a company and its users and is recognized as an important concept along with the acceptance of technology [36]. It occurs after the acceptance phase and is a long term act. Recent studies in marketing and management information claim that the success of a product or service comes not from the consumer's first use but from continuous use [9,26,37–39].

Intention of continuous use is a key concept for maintaining the relationship between users and the company. The satisfaction and expectations based on past experiences play an important role in the formation of it [40]. Bhattacherjee [9] argued that the continued use of customers, rather than initial acceptance, is a key factor in the success of technology-based financial services. Based on that, US banks said that the cost of acquiring new customers and creating new bank accounts and registering them in the system were five times higher than existing customers.

Behavioral loyalty refers to the continued purchase from the same provider, the willingness to recommend to others, and whether or not the relationship continues [41]. Considering repetitive purchasing behavior for a certain period of time for a specific product/service, it can be measured by repeat purchasing behavior, its ratio, and its frequency [42].

Unlike in the past, defensive marketing strategies, such as increasing customer loyalty and retaining existing customers, are more important than aggressive marketing strategies such as attracting new customers and expanding market share. Existing customers with high loyalty frequently purchase large quantities and are less sensitive to competitive price incentives, leading to continuous repurchases and positive word-of-mouth effects, which play a significant role in increasing organization profits through the creation of new customers [43].

Behavioral loyalty, the final determinant of loyalty, begins to develop after attitudinal loyalty, which is a cognitive, emotional, and active loyalty level. It means that customers can continue to buy a particular product or service and to recommend it to their friends. Attitudinal loyalty thus implies the shift of motivated intentions into the behavioral phase and involves efforts to eliminate various

obstacles that hinder the realization of behavior [44]. That is, customers are not easily affected by the various benefits offered by competitors. The effect of this voluntary word of mouth is more credible than the traditional marketing and commercial advertising activities in the company. Firms' optimal input of their resources makes it possible to acquire customers and realize financial income, which is highly useful at the management level [45].

When customers were satisfied, they decided whether to continue using the service or switch to another service. Continuous use of services is considered to be important because users who switch to other services or stop using them can be the center of negative word of mouth [46]. Shamdasani et al. [47] demonstrated that the service quality of online banking has a significant effect on the intention of consumers' continuous use. Zhou's [48] study verified that the system quality factors of smartphone-based mobile payment have a significant effect on the continuous use intention of users by mediating satisfaction.

In the fierce financial market, Internet specialized banks, traditional financial firms, and Fintech companies are making great efforts to increase the loyalty of users and to link them with positive brand image, continuous use, and long-term profit. If consumers become loyal to the use of Internet specialized banks, they will continue to use the financial services. Attitudinal loyalty will therefore act as a leading factor in behavioral loyalty. Based on previous studies, the following research hypotheses are presented.

**Hypothesis 4 (H4)**— *Information of choice attributes in Internet special banks will have a positive influence on intention of continuous use.*

**Hypothesis 5 (H5)**— *Transaction of choice attributes in Internet special banks will have a positive influence on intention of continuous use.*

**Hypothesis 6 (H6)**— *Safety of choice attributes in Internet special banks will have a positive influence on intention of continuous use.*

**Hypothesis 7 (H7)**— *Attitudinal loyalty in Internet special banks will have a positive influence on intention of continuous use.*

*2.5. Gender*

Gender is traditionally the most basic variable of market segmentation, and many studies have revealed differences in gender responses to various marketing stimuli. These differences between men and women have proved their usefulness in predicting attitudes and behavior toward objects in consumer behavior studies [49]. In particular, recent studies have suggested empirical results that demonstrate the difference in perceptions between men and women in the adoption of new information and communication technologies. For example, a study by Nysveen et al. [50] reported gender differences in the motivation for using mobile chat services.

A study by Rodgers and Harris [51] found a difference in perceptions between men and women about the functional aspects of online shopping. Gender has recently been actively used as a moderating variable in the field of technological information, and has been used as a variable to determine an important ratio of products and services in market segmentation [52,53].

The main reason for using demographic characteristics as segmentation variables is not only because they are easy to measure and apply, but they are also closely related to consumer buying behavior. Gender is traditionally the most basic variable of market segmentation in the marketing strategy development phase [50] and can explain consumer behavior in detail. It is also important for marketing managers because it can help predict social trends [54]. Recent studies have shown that female consumers, who are less innovative than men, are sensitive to online information leakage and have a direct impact on acceptance and behavioral intention [55,56].

　　　Therefore, it is expected that there will be a moderating effect of gender according to the choice attributes on Internet specialized banks and it is used as a moderating variable. The following hypotheses are presented to confirm the moderating effects of gender.

**Hypothesis 8-1 (H8-1)**— *There will be a moderating effect of gender when information of choice attributes in Internet specialized banks has a positive impact on attitudinal loyalty.*

**Hypothesis 8-2 (H8-2)**— *There will be a moderating effect of gender when transaction of choice attributes in Internet specialized banks has a positive impact on attitudinal loyalty.*

**Hypothesis 8-3 (H8-3)**— *There will be a moderating effect of gender when safety of choice attributes in Internet specialized banks has a positive impact on attitudinal loyalty.*

**Hypothesis 9-1 (H9-1)**— *There will be a moderating effect of gender when information of choice attributes in Internet specialized banks has a positive impact on intention of continuous use.*

**Hypothesis 9-2 (H9-2)**— *There will be a moderating effect of gender when transaction of choice attributes in Internet specialized banks has a positive impact on intention of continuous use.*

**Hypothesis 9-3 (H9-3)**— *There will be a moderating effect of gender when safety of choice attributes in Internet specialized banks has a positive impact on intention of continuous use.*

## 3. Research Method

### 3.1. Research Model

　　　The research model was constructed with research hypotheses based on relevant literature. It focuses on a perspective to investigate how users of Internet specialized banks evaluate choice attributes (information, transaction, and safety) and form integrated loyalty (attitudinal loyalty and intention of continuous use). It also attempts to identify which of the factors of choice attributes play the most important role in forming integrated loyalty. Furthermore, this study analyzes the possibility of market segmentation by identifying gender differences when choice attributes affect integrated loyalty. The implications for sustainable development activities of the banks will be also presented. Thus, the research model is presented as shown in Figure 1.

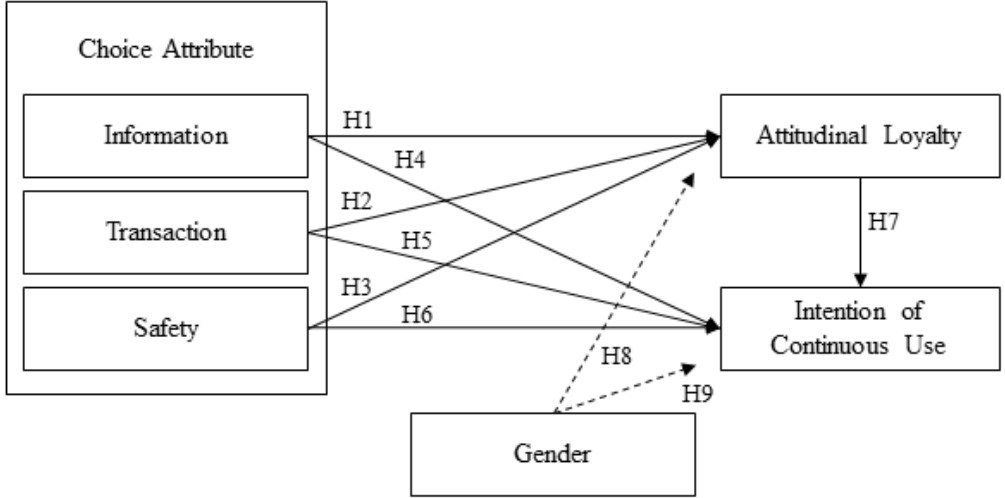

**Figure 1.** Research model.

To empirically validate the proposed research model and hypotheses, we conducted a survey using a questionnaire that includes measurement items for the constructs specified in the proposed research model. The sampling unit were people having an Internet specialized bank account and using its services. People who used the bank formed the sample, which also gives a general idea on the demographics of consumers of the bank in Korea. The survey consisted of two sections, wherein the first section explored the perceptions of the choice attributes, attitudinal loyalty, and intention of continuous use, and the second section comprised of demographic variables. The respondents were asked to rate the perceived variables using a seven-point Likert scale, which ranged from 'strongly disagree' to 'strongly agree' and the demographic variables used nominal scale.

In order to test the proposed relationships among choice attributes (information, transaction, and safety), attitudinal loyalty, and intention of continuous use, structural equation modeling (SEM) was performed using analysis of moment structure (AMOS). SEM can examine the causal relationships among constructs in the model and to test the model against the obtained measurement data to identify how well the proposed model fits the data [57]. It is an appropriate statistical method to examine hypothesized relationships among constructs proposed in this study. Meanwhile, the gender moderating effect analysis of this study used the chi-square difference test of AMOS. The data processing used the SPSS 25.0 and AMOS 21.0 programs.

### 3.2. Measurement Item

A questionnaire approach was used to validate the research mode. For the development and validation of the measurement instrument, the questionnaire items were borrowed from previous studies. The items and constructs used in this study are shown in Table 1. The items used in this study were revised and supplemented to be suitable for Internet specialized banks. Information is defined as accuracy, sufficiency, and understanding of information delivery and information of various product assortments provided by Internet specialized banks. Transaction is defined as simplicity of search and use in financial products, ease of problem solving, quick opening and use, and no difference from ad content. Safety is defined as stable system operation of Internet specialized banks, reliable protection of user's personal information, and safe from hacker intrusion [14]. Attitudinal loyalty is defined as loyal users of Internet specialized banks, recommendation intention to others, and willingness to continue to use due to the best choice [30]. Intention of continuous use is defined as continuing to use the Internet specialized bank in business and no switching to another bank [9].

**Table 1.** Measurement items.

| Construct | Items | Reference |
|---|---|---|
| Information | I1. It has an assortment of products of various kinds of Internet specialized banks. <br> I2. It provides accurate information on Internet specialized bank products. <br> I3. It provides enough information on Internet specialized bank products. <br> I4. The delivery of information about Internet specialized bank products is clear and easy to understand. | |
| Transaction | T1. The process is simple from the search, opening, and use of financial products. <br> T2. The opening and use of desired financial products can be used quickly and accurately. <br> T3. It is easy to solve various problems in the process of using the Internet specialized bank. <br> T4. The information on the financial products and services traded is no different from that displayed and advertised. | [14] |
| Safety | S1. The Internet specialized bank system is operating stably. <br> S2. It well protects personal information of Internet specialized bank users. <br> S3. The Internet specialized bank is safe from outside hackers. | |
| Attitudinal Loyalty | AL1. I will use the Internet specialized bank because it is the best choice for me. <br> AL2. I am a loyal user of the Internet specialized bank. <br> AL3. I am willing to recommend the Internet specialized bank to others. | [30] |
| Intention of Continuous Use | IC1. I will continue to use the Internet specialized bank without stopping. <br> IC2. I will continue to use the Internet specialized bank rather than using another bank. <br> IC3. If possible, I will continue to use the Internet specialized bank. <br> IC4. I will continue to use the Internet specialized bank that I am using. | [9] |

### 3.3. Data Collection

Prior to this study, a preliminary survey was conducted on 10 real Internet specialized bank users to understand the questions and problems in the questionnaire. As a revised and supplemented questionnaire, an online questionnaire was conducted from August to December 2018 for the convenience of the participants in the research on the real users of Internet specialized banks nationwide. The online questionnaires collected data through IT information, electronic device purchasing information community, internet research questionnaire cafe, and SNS. Ten of the respondents who left their contacts as a reward for increasing the survey participation rate were randomly selected and presented a mobile gift. As a result, 250 questionnaires were collected and 215 parts were used for analysis, except 35 questionnaires with missing or unsatisfactory responses.

The sample characteristics are as follows. Among the demographics, concerning gender, there were 132 males (61.4%) and 83 females (38.6%). A total of 61.4% of the respondents are men. This gender difference in the distribution is not surprising given the type of product, as concern for financial transaction has a stronger root in men than in women. The age of the participants is 20s (28.9%) and 30s (42.8%) who actively use financial applications with smartphones. The occupation was office workers (53.5%) and small business person (14.4%). In addition, the most common types of transactions used by Internet specialized banks are duplicate responses, with 152 (70.7%) inquiries and transfers and 98 (45.5%) with credit cards (see Table 2).

**Table 2.** Demographic characteristics.

| Characteristics | Categories | *n* (%) |
|---|---|---|
| Gender | Male | 132 (61.4) |
| | Female | 83 (38.6) |
| Age | Twenties | 62 (28.8) |
| | Thirties | 92 (42.8) |
| | Above Forties | 61 (28.4) |
| Occupation | University student | 24 (11.2) |
| | Postgraduate student | 29 (13.5) |
| | Office workers | 115 (53.5) |
| | Small business person | 31 (14.4) |
| | Other occupations | 16 (7.4) |
| Mainly used transaction type (overlap answer) | Account check/transfer | 152 (70.7) |
| | Deposit/installment savings account | 77 (35.8) |
| | Loan | 32 (14.9) |
| | Overseas remittance | 25 (11.6) |
| | Credit card | 98 (45.5) |

## 4. Data Analysis and Results

### 4.1. Reliability and Validity

Based on the theoretical background, confirmatory factor analysis was conducted to ensure more rigid convergent validity and unidimensionality among the constructs. The results are shown in Table 3. $\chi^2$ = 18.903, df = 70, $p$ = 0.000, $\chi^2$/df = 1.699, GFI = 0.933, AGFI = 0.886, RMR = 0.046, TLI = 0.961, CFI = 0.974, and RMSEA = 0.057, the goodness-of-fit indices of the measurement model, which mean that the model is generally suitable. Furthermore, the results of the reliability test showed that the Cronbach alpha values of all five constructs used in this study exceeded the minimum requirement for reliability of 0.70, which indicated that multiple measurement items were highly reliable for measuring each construct [57]. All the composite reliability is above the threshold of 0.7, indicating that the measurement model has good reliability of the constructs [58]. Convergent validity was also examined with the factor loadings in the measurement model. The mean variance extraction value (AVE) is above

the standard value of 0.5, which suggests that the measurement model has convergent validity [57] (see Table 3).

**Table 3.** Reliability and validity for the measurement model.

| Construct | Item | Standardized Estimate | S.E. | t-value | Composite Reliability | Cronbach's $\alpha$ | AVE |
|---|---|---|---|---|---|---|---|
| Information | I1 | 0.737 | - | - | 0.870 | 0.849 | 0.692 |
| | I3 | 0.823 | 0.101 | 11.497 | | | |
| | I4 | 0.874 | 0.105 | 11.705 | | | |
| Transaction | T2 | 0.798 | - | - | 0.878 | 0.849 | 0.708 |
| | T3 | 0.929 | 0.084 | 14.350 | | | |
| | T4 | 0.733 | 0.085 | 12.134 | | | |
| Safety | S1 | 0.741 | - | - | 0.860 | 0.838 | 0.673 |
| | S2 | 0.899 | 0.090 | 12.219 | | | |
| | S3 | 0.764 | 0.092 | 11.520 | | | |
| Attitudinal Loyalty | AL1 | 0.720 | - | - | 0.880 | 0.854 | 0.712 |
| | AL2 | 0.930 | 0.095 | 13.678 | | | |
| | AL3 | 0.816 | 0.092 | 13.108 | | | |
| Intention of Continuous Use | IC1 | 0.819 | - | - | 0.898 | 0.898 | 0.748 |
| | IC2 | 0.934 | 0.073 | 15.433 | | | |
| | IC3 | 0.770 | 0.078 | 13.148 | | | |

$\chi^2$ = 118.903, df = 70, p = 0.000, GFI = 0.933, AGFI = 0.886, RMR = 0.046, TLI = 0.961, CFI = 0.974, RMSEA = 0.057.

Through the results of confirmatory factor analysis, the unidimensionality of the constructs is identified. Correlation analysis was conducted to confirm the direction and degree of relationship between the constructs. In Table 4, the positive correlation between the variables used in this study was significant at the significance level of 0.01. This shows that the better the choice attributes, the higher the integrated loyalty (attitudinal loyalty and intention of continuous use). Fornell and Larcker [59] suggested a more robust method of measuring discriminant validity, in which a correlation between two constructs should be lower than the squared root of the AVE value for any of the two constructs. Since the values of the square root of AVEs are more than 0.7 and higher than the correlation coefficient values, the discriminant validity can be satisfied (see Table 4).

**Table 4.** Discriminant validity.

| Title 1 | (1) | (2) | (3) | (4) | (5) |
|---|---|---|---|---|---|
| (1) Information | **0.832** | | | | |
| (2) Transaction | 0.304** | **0.841** | | | |
| (3) Safety | 0.230** | 0.283** | **0.821** | | |
| (4) Attitudinal Loyalty | 0.391** | 0.470** | 0.377** | **0.844** | |
| (5) Intention of Continuous Use | 0.406** | 0.376** | 0.308** | 0.520** | **0.865** |

The diagonal bold is the AVE square root value. **: $p < 0.01$.

### 4.2. Structural Model Analysis

In this study, the structural equation model was used to test the hypotheses. $\chi^2$ = 117.461, df = 70, $p$ = 0.000, $\chi^2$/df = 1.678, GFI = 0.937, AGFI = 0.892, RMR = 0.044, TLI = 0.962, CFI = 0.975, and RMSEA = 0.056, the goodness-of-fit indices of the path model, which mean that the model is generally suitable. The structural results of the proposed model are depicted in Figure 2. Path analysis results of this study are shown in Table 5. The results of the moderating effect of gender are presented in Table 6.

First, the hypothesis that information of choice attributes in Internet specialized banks will have a positive effect on attitudinal loyalty (H1) was adopted with a standard coefficient of 0.230 and t = 3.258

($p = 0.001$). It is shown that the information provided by the banks can lead to high levels of attitudinal loyalty if detailed, accurate, and sufficient.

Second, the hypothesis that transaction of choice attributes in Internet specialized banks will have a positive effect on attitudinal loyalty (H2) was adopted with a standard coefficient of 0.338 and t = 4.888 ($p = 0.000$). Transaction was measured to be the highest value among the choice attributes. In order to increase attitudinal loyalty of the banks, the procedures and functional conveniences required by users in using financial products can be the most important. These characteristics can emphasize the advantages of the banks specializing in simple and convenient than the existing Internet banking or mobile banking.

Third, the hypothesis that safety of choice attributes in Internet specialized banks will have a positive effect on attitudinal loyalty (H3) was adopted with a standard coefficient of 0.214 and t = 3.157 ($p = 0.002$). The higher the level of safety and speed of using the bank systems, the more it can lead to attitudinal loyalty.

Fourth, the hypothesis that information of choice attributes in Internet specialized banks will have a positive effect on intention of continuous use (H4) was adopted with a standard coefficient of 0.231 and t = 2.889 ($p = 0.004$). It shows that information provided by the banks can lead to continued use if detailed, accurate, and sufficient.

Fifth, the hypothesis that transaction of choice attributes in Internet specialized banks will have a positive effect on intention of continuous use (H5) was rejected with a standard coefficient of 0.135 and t = 1.704 ($p = 0.088$). Transaction of the banks was found to have no influence on attitudinal loyalty. This shows that users of the banks do not value transaction attributes as important in shaping attitudinal loyalty.

Sixth, the hypothesis that safety of choice attributes in Internet specialized banks will have a positive effect on intention of continuous use (H6) was adopted with a standard coefficient of 0.149 and t = 1.991 ($p = 0.047$). It is shown that the safety and speed of the bank systems can lead to behavioral loyalty such as continuous use or recommendation.

Seventh, the hypothesis that attitudinal loyalty of Internet specialized banks will have a positive effect on intention of continuous use (H7) was adopted with a standard coefficient of 0.389 and t = 4.154 ($p = 0.000$). This shows that the attitudinal loyalty of the banks can lead to continuous use, which is behavioral loyalty.

Eighth, when choice attributes (information, transaction, and safety) had a positive effect on attitudinal loyalty in Internet specialized banks, we analyzed whether there is a difference in the effects of gender (H8-1–H8-3). In Table 6, the chi-square difference test of AMOS showed that the $\triangle\chi^2$ (1) values were lower than 3.84 in the comparison between the free model $\chi^2$ and constraint model $\chi^2$ values, suggesting that the hypotheses (H8-1–H8-3) are rejected at the significance level of 0.05. It shows that there is no gender moderating effect between the choice attributes and attitudinal loyalty of the banks. However, there is a difference in the value of each path coefficient of male and female. Male path values were a little high when the two variables of choice attributes had a positive effect on attitudinal loyalty.

Ninth, when choice attributes (information, transaction, and safety) had a positive effect on intention of continuous use in Internet specialized banks, we analyzed whether there is a difference in the effects of gender (H9-1–H9-3). In Table 6, the chi-square difference test of AMOS showed that the $\triangle\chi^2$ (1) values were lower than 3.84 in the comparison between the free model $\chi^2$ and constraint model $\chi^2$ values, suggesting that the hypotheses (H9-1–H9-3) are rejected at the significance level of 0.05. This shows that there is no moderating effect of gender between the choice attributes and intention of continuous use in the banks. However, there is a difference in the value of each path coefficient of male and female. Male path values were a little high when choice attributes had a positive effect on intention of continuous use.

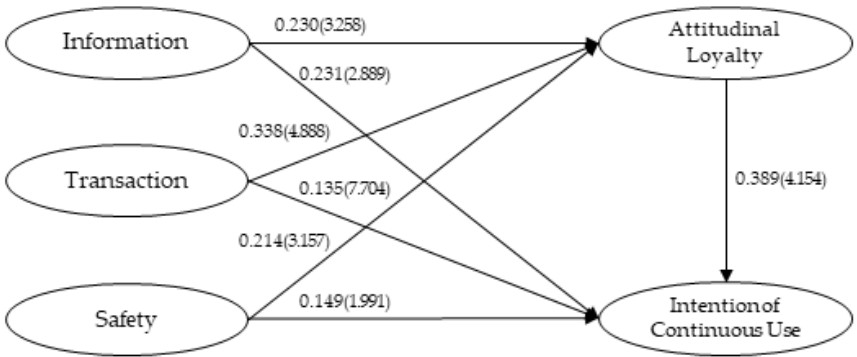

**Figure 2.** Path model.

**Table 5.** Results of hypothesis testing.

| | Hypothesized Path | Std. Estimate | S.E. | t-value | *p*-value | Results |
|---|---|---|---|---|---|---|
| H1 | Information → Attitudinal Loyalty | 0.230 | 0.071 | 3.258 | 0.001 | Accepted |
| H2 | Transaction → Attitudinal Loyalty | 0.338 | 0.069 | 4.888 | 0.000 | Accepted |
| H3 | Safety → Attitudinal Loyalty | 0.214 | 0.068 | 3.157 | 0.002 | Accepted |
| H4 | Information → Intention of Continuous Use | 0.231 | 0.080 | 2.889 | 0.004 | Accepted |
| H5 | Transaction → Intention of Continuous Use | 0.135 | 0.079 | 1.704 | 0.088 | Rejected |
| H6 | Safety → Intention of Continuous Use | 0.149 | 0.075 | 1.991 | 0.047 | Accepted |
| H7 | Attitudinal Loyalty → Intention of Continuous Use | 0.389 | 0.094 | 4.154 | 0.000 | Accepted |

$\chi^2$ = 117.461, df = 70, p = 0.000, GFI = 0.937, AGFI = 0.892, RMR = 0.044, TLI = 0.962, CFI = 0.975, RMSEA = 0.056.

**Table 6.** Moderating effect testing.

| | Hypothesized Path | Male (*n* = 132) | | Female (*n* = 83) | | Free Model | Constrained Model |
|---|---|---|---|---|---|---|---|
| | | Std. Estimate | t-value | Std. Estimate | t-value | | |
| H8-1 | Information → Attitudinal Loyalty | 0.263 | 2.727 | 0.184 | 1.851 | $\chi^2(140)$ = 201.782 | $\chi^2(141)$ = 202.095 |
| | Chi-square difference test: $\triangle\chi^2(1)$ = 0.313 (n.s.) | | | | | | |
| H8-2 | Transaction → Attitudinal Loyalty | 0.261 | 2.902 | 0.435 | 4.093 | $\chi^2(140)$ = 201.782 | $\chi^2(141)$ = 203.338 |
| | Chi-square difference test: $\triangle\chi^2(1)$ = 1.556 (n.s.) | | | | | | |
| H8-3 | Safety → Attitudinal Loyalty | 0.225 | 2.676 | 0.220 | 1.892 | $\chi^2(140)$ = 201.782 | $\chi^2(141)$ = 201.783 |
| | Chi-square difference test: $\triangle\chi^2(1)$ = 0.001 (n.s.) | | | | | | |
| H9-1 | Information → Intention of Continuous Use | 0.257 | 2.231 | 0.179 | 1.664 | $\chi^2(140)$ = 201.782 | $\chi^2(141)$ = 202.049 |
| | Chi-square difference test: $\triangle\chi^2(1)$ = 0.267 (n.s.) | | | | | | |
| H9-2 | Transaction → Intention of Continuous Use | 0.165 | 1.762 | 0.155 | 1.233 | $\chi^2(140)$ = 201.782 | $\chi^2(141)$ = 201.786 |
| | Chi-square difference test: $\triangle\chi^2(1)$ = 0.004 (n.s.) | | | | | | |
| H9-3 | Safety → Intention of Continuous Use | 0.196 | 2.231 | 0.023 | 0.183 | $\chi^2(140)$ = 201.782 | $\chi^2(141)$ = 203.048 |
| | Chi-square difference test: $\triangle\chi^2(1)$ = 1.266 (n.s.) $\triangle\chi^2(1)$ > 6.63: p < 0.01; $\triangle\chi^2(1)$ > 3.84: p < 0.05 | | | | | | |

## 5. Conclusions

Internet specialized banks are entering the growth phase after the introduction period in Korea. If many financial consumers do not feel the difference between Internet specialized banks and Internet banking, this may be due to the failure of the Internet specialized banks' marketing strategy. It would be necessary for Internet specialized banks to impress financial service users with a unique differentiation strategy that is different from existing banks' Internet banking. Therefore, it is a very important task to find a way to improve attitudinal loyalty and intention of continuous use in Internet specialized banks. This study suggests the choice attributes (information, transaction, and safety) as a means of improving attitudinal loyalty and intention of continuous use in Internet specialized banks. We also examined whether there is a difference between men and women when the choice attributes affect attitudinal loyalty and intention of continuous use.

The results of the empirical analysis are summarized as follows. First, the choice attributes (information, transaction, and safety) of Internet specialized banks have a positive effect on attitudinal loyalty. Second, the choice attributes (information and safety) of Internet specialized banks have a positive effect on intention of continuous use. Third, attitudinal loyalty has a positive effect on intention of continuous use. Fourth, there is no difference between men and women when the choice attributes (information, transaction, and safety) have a positive effect on attitudinal loyalty and intention of continuous use.

Based on the results of this analysis, the academic implications of theoretical area can be presented as follows. First, while previous studies of Internet specialized banks have focused on technology acceptance model or innovation acceptance model [1,2], this study approaches an integrated loyalty model. Although it is meaningful to apply the technology acceptance model in connection with the introduction of Internet specialized banks, it is necessary to apply the loyalty model because the success of the bank depends on attitudinal and behavioral loyalty. In this study, attitude loyalty has a positive effect on intention of continuous use in behavioral loyalty. A composite of the two variables can lead to actual behavior [28]. Increasing the likelihood of Internet specialized banks' success will depend on their ability to accurately predict the actual behavior of consumers. Thus, there is a need to strengthen the role of integrated loyalty in the Internet specialized bank market. In this market, research on integrated loyalty will be activated.

Second, Ahn and Lee [13] analyzed the relationship between perceived value (economic value, convenience value, and emotional value) and usage intention in the research on Internet specialized banks as behavioral models. Convenience value was found to have the most influence on usage intention. They explained convenience value as service convenience (information, accessibility, and ease of transaction, and save time and effort) of Internet specialized banks. Today, financial consumers can easily find differences in customer service between banks through comparison sites, so the choice of service is important. In this regard, the model of the relationship between choice attributes (information, transaction, and safety), attitudinal loyalty, and intention of continuous use can be useful in this field of research. Therefore, users' needs for Internet specialized banks can be considered as choice attributes, and a differentiated strategy on attributes will be meaningful.

Third, in this study, we examined the moderating effects of gender that were rarely investigated in previous studies on Internet specialized banks. Chiu et al. [55] found that female consumers, who are less innovative than men, are sensitive to information leakage online and have a direct impact on acceptance and behavioral intention. In this study, however, there was no gender difference when the choice attribute (information, transaction, and safety) affected attitudinal loyalty and intention of continuous use. This shows that there is no difference in the needs between male and female users for the choice attributes of the Internet specialized bank. In this study, the moderating effect of gender is not statistically significant, but there is a subtle difference between men and women, so it is necessary to approach the CRM marketing strategy. CRM marketing based on big data is found to be useful in overcoming the limitations of existing segmented market strategies. If Internet specialized banks can

provide services that reflect the subtle differences between men and women, the banks will be able to achieve meaningful performance.

The practical implications can be provided as follows. First, the choice attributes (information, transaction, and security) have a positive effect on attitudinal loyalty and intention of continuous use, and these variables can be the main influencing factors to enhance the integrated loyalty of Internet specialized banks. Among the choice attributes, transaction shows more influence on attitudinal loyalty than information and safety do. On the other hand, safety is found to have the least impact. It can be seen that the ease of searching, opening, using, fast processing, and trust in advertisement promotion play an important role in improving attitudinal loyalty in financial product transactions of Internet specialized banks.

Meanwhile, among the choice attributes, information shows more influence on intention of continuous use than transaction and safety do. On the other hand, transaction is found to have the least impact. It can be seen that the provision of various financial product information and the accuracy, sufficiency, and persuasive delivery of information in Internet banks specialized play an important role in improving intention of continuous use. Therefore, in order to enhance the attitudinal loyalty and continuous use intention of Internet specialized banks, it is necessary to develop and manage the choice attributes of financial services from the perspective of users.

Second, a positive relationship is found between attitudinal loyalty and intention of continuous use. This is consistent with the results of previous studies [30]. It is argued that the success of a product or service comes not from the consumer's first use but from continuous use [9,26]. In this study, attitudinal loyalty has more influence on intention of continuous use than the choice attributes do. Attitudinal loyalty is found to be a critical factor in intention of continuous use. Thus, increasing the attitudinal loyalty of Internet specialized bank users may lead to the successful settlement of the banks.

Third, it is confirmed that the relationship between the choice attribute, attitudinal loyalty, and intention of continuous use can be the model necessary for the success of Internet specialized banks. If an Internet specialized bank is well developed and managed in terms of the users' perspectives, it will improve its loyalty and financial performance. Since companies are social beings, they need a responsible attitude to solve the problems of climate change, environmental protection, and resource depletion that are required at the present time. Internet specialized banks should also play a role in solving these problems. If the public image of the banks improves, customer loyalty will become stronger. Therefore, there is a need to promote more ethical codes of Internet specialized banks such as environmental protection for the sustainable development of future generations and elimination of alienation of digital financial services for the elderly.

Fourth, although gender differences were not statistically significant when the choice attributes affected integrated loyalty (attitudinal loyalty and intention of continuous use), it is interesting to note that there are very subtle differences between men and women. Among the choice attributes affecting attitudinal loyalty, men valued information and women valued transaction more importantly. Additionally, in the choice attributes affecting intention of continuous use, it was found that men valued three factors more importantly than women did. Thus, the market segmentation strategy by gender needs more careful approach in Internet specialized bank transactions.

## 6. Limitations and Future Research

Implications were suggested based on the results of the analysis, but they have the following limitations. First, in this study, data collection was done only once, and the number of questionnaires used for empirical analysis was rather difficult to generalize. Further surveys and analyses are needed for more specific and better results in the future. Meanwhile, the proportion of women in this sample is much lower than that of men. Since women are more active in online transactions than men presently, future research will need to address this.

Second, this study is about Internet specialized banks, but if the research model used includes the service quality evaluation on the choice attributes, more meaningful implications can be suggested.

The choice attributes of a product or service can be helpful in a differentiation strategy. In order to highlight the differentiation strategy from the existing Internet banking, it is necessary to supplement this part in the future research in order to provide the uniqueness of the Internet specialized bank service. In general, service quality, satisfaction, and profitability are intimately connected, and high levels of quality bring about high levels of satisfaction, which in turn have a positive impact on the performance of companies [15]. Thus, the quality-satisfaction-profitability model needs to be incorporated into future research on Internet specialized banks.

Third, the study could be further developed by applying the constructs used in this study to products and services other than Internet specialized banks, or by developing various leading and trailing variables. In general, it is known that the choice attribute of a product or service should reflect the needs of consumers to increase the likelihood of purchase. In this study, only three optional attributes were used, but future studies need to include additional attributes.

Fourth, in this study, this study proposed a method of activating the Internet specialized bank using a model of choice attribute-attitudinal loyalty-intention of continuous use. Since this model does not provide sufficient explanation, future research will need to expand the model.

Fifth, OECD (Organization for Economic Cooperation and Development) studies of household behavior showed that women are more likely than men to buy recyclable, eco-labeled, and energy-efficient products. Unfortunately, sustainable production is not following directly from higher levels of sustainable consumption by women [60]. Thus, in the future, research on Internet specialized banks will need to investigate the role of gender in sustainability.

**Author Contributions:** J.-H.J. collected and analyzed the data and prepared the draft, and J.-I.S. organized the research design and finalized the paper.

**Funding:** This research received no external funding.

**Acknowledgments:** This research was supported by Gyeongnam National University of Science and Technology and University of Ulsan.

**Conflicts of Interest:** The authors declare no conflict of interest.

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
