# Peer review of "The Effect of Choice Attributes of Internet Specialized Banks on Integrated Loyalty: The Moderating Effect of Gender"

_sustainability, doi:10.3390/su11247063_

Round 1
Reviewer 1 Report
In Introduction section, this paper needs to identify the limitations of the results of prior studies on the topic Among choice attributes, why three factors such as information, transaction, and safety is more important than any other factors in prior studies. This paper needs to specify the reason and logic to choose three factors. In research model section, this paper needs to theorectically specify the logic of research model in comparison with prior studies on the topic In data collection section, Table 2 has some error to describe Gender (for example, Male 132 (61.4%), Female 33 (33.6%). It will be revised to Male 132 (80.0%), Female 33 (20.0%). I think H5 is rejected, because of P-value (at least 0.05 significance level) In conclusion section, this paper needs to divide the conclusion into two parts . One is academic implications on the academic contribution of theorectical area, the other is practical implications from the viewpoint of manager of Internet specialized banks.
Author Response
Point 1: In Introduction section, this paper needs to identify the limitations of the results of prior studies on the topic Among choice attributes, why three factors such as information, transaction, and safety is more important than any other factors in prior studies. This paper needs to specify the reason and logic to choose three factors. In research model section, this paper needs to theorectically specify the logic of research model in comparison with prior studies on the topic.
Response 1: Additional explanations are given on pages 4 and 7.
Point 2: In data collection section, Table 2 has some error to describe Gender (for example, Male 132 (61.4%), Female 33 (33.6%). It will be revised to Male 132 (80.0%), Female 33 (20.0%).
Response 2: We correct what you pointed out on page 9.
Point 3: I think H5 is rejected, because of P-value (at least 0.05 significance level)
Response 3: We correct what you pointed out on page 11.
Point 4: In conclusion section, this paper needs to divide the conclusion into two parts. One is academic implications on the academic contribution of theorectical area, the other is practical implications from the viewpoint of manager of Internet specialized banks.
Response 4: We correct what you pointed out on page 13 and 14.
Reviewer 2 Report
I definitely believe that the topic is worth investigating and the article is interesting. The analysis has several flaws, which make it unsuitable for publication in its current form. I provide more details below, which can hopefully of guidance to the Authors. I wish them good luck with their work.
Our suggestions:
The Title, Introduction, Literature Review, Methodology, Empirical Results, and Conclusions should be rearranged to determine the notion of sustainability in their research. The introduction – and thereby the analysis on the whole – suffers from a confusion of concepts. The title of Journal is “Sustainability”. The Authors don’t explain how their research is related to the sustainability. The term “sustainability” doesn’t appear in the text.
I feel there are other Mdpi journals that may better suit the scope. So, I recommend a transfer this interesting article to another Mdpi journal.
The Authors do not discuss limitations of the method they applied, nor do they discuss alternative concepts used in their field of research. The criteria for selecting respondents for the research need to be described in the Methodology section.
Hope this suggestions may help Authors to improve their work.
Author Response
Point 1: The Title, Introduction, Literature Review, Methodology, Empirical Results, and Conclusions should be rearranged to determine the notion of sustainability in their research. The introduction – and thereby the analysis on the whole – suffers from a confusion of concepts. The title of Journal is “Sustainability”. The Authors don’t explain how their research is related to the sustainability. The term “sustainability” doesn’t appear in the text.
Response 1: The notion of sustainability is presented in addition to abstract, introduction, and conclusion.
Point 2: The Authors do not discuss limitations of the method they applied, nor do they discuss alternative concepts used in their field of research. The criteria for selecting respondents for the research need to be described in the Methodology section.
Response 2: What you pointed out is presented in limitation of conclusions.
Reviewer 3 Report
This is an interesting and topical study of internet banking. It uses data from a questionnaire on attitudes to internet banking to determine which factors affect loyalty, including gender differences. Overall loyalty is driven by information, transaction and safety features. I have the following comments:
-In the introduction, a list of the contributions to the literature could be added. -The survey that was conducted needs a bit more information about added, especially on how it was made a random selection. How were the respondents identified by the authors and approached and what measures were taken to ensure a random sample.
-The study uses correlation coefficients and measures of significance, the authors could add some discussion on why they used this approach rather than the multivariate regression analysis.
Author Response
Point 1: In the introduction, a list of the contributions to the literature could be added.
Response 1: What you pointed out is provided in the introduction.
Point 2: The survey that was conducted needs a bit more information about added, especially on how it was made a random selection. How were the respondents identified by the authors and approached and what measures were taken to ensure a random sample.
Response 2: What you pointed out is provided on page 8.
Point 3: The study uses correlation coefficients and measures of significance, the authors could add some discussion on why they used this approach rather than the multivariate regression analysis.
Response 3: What you pointed out is provided on page 9.
Round 2
Reviewer 2 Report
1. I appreciate the improvements introduced into the article, particularly the added text in introduction. But the whole paper needs to be much more streamlined and focused on the influence of Effect the Gender upon sustainable development. Furthermore, some parts of the article are particularly confusing on the matter.
It could be useful to explain how the research influence on the sustainability in the specialized banks in South Korea.
Fundamentally the Chapter 2. (Literature Review and Hypothesis) and Chapter 5. (Conclusions) fail to address the issue of sustainable development.
2. The criteria for selecting respondents for the research need to be described in the Methodology section. A selection of the sample is very important in the case of Internet specialized banks.
Hope these suggestions may help Authors to improve their work.
Author Response
Point 1: But the whole paper needs to be much more streamlined and focused on the influence of Effect the Gender upon sustainable development. Furthermore, some parts of the article are particularly confusing on the matter. Fundamentally the Chapter 2. (Literature Review and Hypothesis) and Chapter 5. (Conclusions) fail to address the issue of sustainable development.
Response 1: We present the revised content of sustainable development in the Introduction, Literature Review and Hypothesis, Research Model, and Conclusion.
Point2: The criteria for selecting respondents for the research need to be described in the Methodology section. A selection of the sample is very important in the case of Internet specialized banks.
Response 2: We present the criteria for selecting respondents in the Data Collection part.